# Characterization of Tetrathionate Hydrolase from Acidothermophilic Sulfur-Oxidizing Archaeon *Metallosphaera cuprina* Ar-4

**DOI:** 10.3390/ijms26031338

**Published:** 2025-02-05

**Authors:** Pei Wang, Liang-Zhi Li, Li-Jun Liu, Ya-Ling Qin, Xiu-Tong Li, Hua-Qun Yin, De-Feng Li, Shuang-Jiang Liu, Cheng-Ying Jiang

**Affiliations:** 1State Key Laboratory of Microbial Resources, Institute of Microbiology, Chinese Academy of Sciences, Beijing 100101, China; pei.wang@siat.ac.cn (P.W.); qinyaling19@mails.ucas.ac.cn (Y.-L.Q.); lixiutong18@mails.ucas.ac.cn (X.-T.L.); lidefeng@im.ac.cn (D.-F.L.); 2University of Chinese Academy of Sciences, Beijing 100049, China; 3School of Minerals Processing and Bioengineering, Key Laboratory of Biometallurgy of Ministry of Education, Central South University, Changsha 410083, China; 205601006@csu.edu.cn (L.-Z.L.); yinhuaqun_cs@sina.com (H.-Q.Y.); 4School of Basic Medical Science, Xi’an Medical University, Xi’an 710021, China; liulijun@xiyi.edu.cn

**Keywords:** *Metallosphaera cuprina*, tetrathionate hydrolase, localization, kinetics of enzyme

## Abstract

Tetrathionate hydrolase (TTH) is a key enzyme for the oxidation of reduced inorganic sulfur compounds (RISCs) with the S_4_I pathway, which is distributed in autotrophic or facultative autotrophic sulfur-oxidizing bacteria and archaea. In this study, the enzyme TTH_Mc_ from the acidothermophilic archaeon *Metallosphaera cuprina* Ar-4^T^, encoded by *mcup*_1281 and belonging to the pyrroloquinoline quinone (PQQ) family, has been shown to possess tetrathionate hydrolysis activity. The molecular mass of the single subunit of TTH_Mc_ was determined to be 57 kDa. TTH_Mc_ is proved to be located in the cytoplasm, periplasmic space, and membrane, and the activity of them accounted for 72.3%, 24.0%, and 3.7% of the total activity. Optimal activity was observed at temperatures above 95 °C and pH 6.0, and the kinetic constants K_m_ and V_max_ were 0.35 mmol/L and 86.3 μmol/L, respectively. The presence of 0.01 mol/L Mg^2+^ enhances the activity of TTH_Mc_, while 0.01 mol/L Ca^2+^ inhibits its activity. The hydrolysis of tetrathionate (TT) by TTH_Mc_ results in the production of thiosulfate, pentathionate, and hexathionate. This study represents the first description of TTH in the genus *Metallosphaera*, providing new theoretical insights into the study of sulfur-oxidizing proteins in acidothermophilic archaea.

## 1. Introduction

Reduced inorganic sulfur compounds (RISCs) are widely distributed in hot springs, sedimentary mud, volcanic craters, and acidic wastewater [1,2,3]. The dissimilatory sulfur oxidation of RISC is a significant geochemical process, which is carried out exclusively by prokaryotes, and the energy generated during this process is coupled with photosynthesis and respiratory processes [4,5]. *Metallosphaera cuprina*, a member of the *Sulfolobales* order within the archaeal domain, was isolated from the sediment of an acidic hot spring in Tengchong County, Yunnan Province, China. This organism thrives at temperatures ranging from 55 to 75 °C and a pH range of 2.5–5.5. *M. cuprina* is a facultative chemoautotroph that oxidizes S^0^, pyrite, and tetrathionate to produce sulfuric acid under aerobic conditions while also capable of chemoheterotrophically growing with yeast extract [6,7,8]. The members of *Sulfolobales* show great potential for bioleaching [9,10].

Archaea with sulfur-oxidizing function have been widely observed in the *Sulfolobales* order [11]. Among the *Sulfolobales* members, *Metallosphaera* [12], *Acidianus* [13], and *Sulfolobus* [14] are representative sulfur-oxidizing archaea found in high-temperature and acidic environments. These species are capable of metabolizing various sulfur compounds and exhibiting heterotrophic and other physiological characteristics that enhance their adaptability to local environments. Previous research on thermophilic archaea has primarily focused on studying the enzymes involved in the sulfur oxidation process, resulting in the discovery of a series of enzymes, such as Sulfur Oxygenase Reductase (SOR), Sulfide:Quinone Oxidoreductase (SQR), Sulfite-acceptor Oxidoreductases (SAOR), Thiosulfate–Quinone Oxidoreductase (TQO), and Tetrathionate Hydrolase (TTH) in *Acidianus brierleyi* (*Ac. brierleyi*) and *Acidianus ambivalens* (*Ac. ambivalens*) [11,15]. 

Meanwhile, there have been numerous studies on the well-established sulfur-oxidizing systems from bacteria [16,17,18] such as *Acidithiobacillus ferrooxidans* (*At*. *ferrooxidans*), *Acidithiobacillus*
*caldus* (*At*. *caldus*), and *Acidiphilium acidophilum* (*Ap. acidophilum*). The enzyme-catalyzed reactions in these bacteria are most suitable at pH levels between 2.0 and 4.0. Studies have reported that the TTH of *At. ferrooxidans* and *At. caldus* are both located in the periplasmic space of cells [19,20]. The TTH of *At. ferrooxidans* hydrolyzes S_4_O_6_^2−^ into S_2_O_3_^2−^, S^0^, and SO_4_^2−^, while the TTH of *At. caldus* produces S_2_O_3_^2−^, S_5_O_6_^2−^, and SO_4_^2−^ [19,20,21,22]. In *Thermithiobacillus tepidarius* (former *Thiobacillus tepidarius*), the TTH was supposed to be located either on or inside the cytoplasmic membrane, and oxidize tetrathionate to S^0^, SO_3_^2−^, and other polythionates [23]. In archaea, the TTH of *Ac. ambivalens* is located in the pseudo-periplasmic space of the cell and is connected to the outer membrane of the cell membrane, displaying optimal activity at pH 1.0 and 95 °C or above [15]. The TTH of *Acidianus hospitalis* (*Ac. hospitalis*) is found in extracellular proteins [24,25]. When exposed to ultraviolet rays and mitogen C stress, the extracellular secretion of *Ac. hospitalis* TTH was promoted [24].

Kanao et al. utilized Western blot to examine the regulation of TTH in *Acidithiobacillus thiooxidans* in the presence of different substrates. The expression of TTH was significantly higher when cultured with S_4_O_6_^2−^ as substrate compared to that with S^0^ and S_2_O_3_^2−^ as substrates. *At. ferrooxidans* cultured with S_4_O_6_^2−^ expressed the *tth* gene while it did not express it when iron was used as the substrate [26,27]. Through transcriptomic and proteomic analyses, it was observed that *At. caldus* expressed the TTH gene when cultured with S_4_O_6_^2−^ as the substrate, but did not when cultured with S^0^ alone [20]. The results from RT-qPCR indicated that the expression level of TTH with S_4_O_6_^2−^ as the substrate in *At. caldus* was about 200 times higher than that with S^0^ as the substrate [28]. Protze’s Northern blot experiment demonstrated that when *Ac. ambivalens* was cultured with S_4_O_6_^2−^ as the substrate, the transcription level of TTH RNA was significantly higher compared to when cultured with S^0^ as the substrate [15]. Furthermore, TTH activity with cyanolysis could only be detected when S_4_O_6_^2−^ was used as the substrate [15]. Research on TTH in archaea primarily focused on the genus *Acidianus*. Although high homologous TTH gene sequences have also been identified in the genome of the genus *Metallosphere*, they have not been identified in the proteome or extensively studied in the field of biochemistry and molecular biology.

In this study, we present evidence that *M. cuprina* Ar-4 cultured with S_4_O_6_^2−^ displays a higher TTH expression level compared with S^0^ as the sole sulfur source. Additionally, we demonstrate that the TTH protein of *M. cuprina* Ar-4 is mainly present in cytoplasm and periplasm and exhibits an optimum activity at pH of 6.0, distinguishing it from other TTH proteins, which show their activity at pH 1.0–4.0 and exist solely in the periplasm or cytoplasm or extracellular.

## 2. Results

### 2.1. Gene Sequence Analysis and Structure Prediction of Tetrathionate Hydrolase of M. cuprina

Based on the genome analysis, it is supposed that *mcup*_*1281* of *M. cuprina* Ar-4 might encode a homologous protein of the TTH1 of *Ac. ambivlaens*. Further analysis was conducted using the ExPASy ProtParam tool and it was determined that *mcup*_*1281* encoded an open reading frame (ORF) consisting of 537 amino acid residues. The protein did not contain any disulfide bonds and only had one cysteine residue in the sequence. The presumed molecular weight was around 57.4 kDa. Further analysis of the protein’s secondary structure predicted the presence of a transmembrane region. These findings suggest that Mcup_1281 is likely located in the extracellular space and is associated with the cell membrane.

AlphaFold2 was used to predict the three-dimensional model. The Mcup_1281 monomer structure appeared to be mainly consisted of beta strands and coil structures and exhibited an eight-bladed β-propeller motif with each blade (I–VIII) consisting of an average of four antiparallel β-strands. Five α-helices were located outside the β-propeller region. Two of these α-helices (α2–α3) were suggested to participate in dimerization [19]. No c-type cytochrome domain was identified in Mcup_1281, unlike the nitrite reductase (Nir PDB 1QKS with RMSD 2.9 Å). The tryptophan residues in the Mcup_1281 were predominantly located within the beta strands region similar to the TTH of *At. ferrooxidans* (PDB 6L8A with RMSD 1.5 Å) (Appendix A).

### 2.2. The TTH Gene Is Expressed Both in Tetrathionate and S^0^ Grown Cells of M. cuprina

To analyze the transcriptional expression of *mcup*_*1281*, total RNA from cells cultured in different mediums (5 g/L S^0^, 5 mmol/L K_2_S_4_O_6_, or 2 g/L yeast extract) was extracted, respectively. The transcription levels of *mcup*_*1281* under different culture conditions were detected via RT-qPCR, and the 16S rRNA gene was used as the reference gene. The results showed that the transcription level of *mcup*_*1281* increased by 5.4-fold and 4-fold when K_2_S_4_O_6_ and S^0^ were used as the energy sources compared to the control group (yeast extract as an energy source) (Figure 1). Therefore, we propose that Mcup_1281 plays an important role in the sulfur metabolism of *M. cuprina*, and define the Mcup_1281 as TTH_Mc_.

### 2.3. Preliminary Study on TTH Activity of M. cuprina

The different cell lysates of *Metallosphaera cuprina* Ar-4 grown aerobically using three different mediums (Basic Salts Medium-added 5 g/L S^0^, 5 mmol/L K_2_S_4_O_6_, or 2 g/L yeast extract separately [6,14], see details in the methods, Section 4.1) were, respectively, detected by the continuous enzyme activity assay method (detailed as Section 4.6) [29]. One unit enzyme activity (U) was defined as the increase in absorbance at 290 nm per minute (∆A_290_ min^−1^) of cell lysates. When using 1 mmol/L K_2_S_4_O_6_ as the substrate and incubating reaction mixture at 65 °C and pH 3.0, the specific enzyme activity (activity of per mg protein) was 0.60 U/mg protein for the cells grown on 5 mmol/L K_2_S_4_O_6_ and 0.29 U/mg protein for the cells grown on 5 g/L S^0^, respectively. The TTH_Mc_ activity with K_2_S_4_O_6_ as the energy source was almost two times higher than that with S^0^. However, the TTH_Mc_ activity was not detected in the cell lysate cultured in yeast extract (Figure 2). The results were in line with the transcription results of Figure 1.

### 2.4. Tetrathionate Hydrolase Primarily Located in Cytoplasm and Periplasm

The discontinuous method (see Section 4.6.2) [30] was used to detect the tetrathionate hydrolase (TTH) activity of Mcup_1281 in various cell fractions, including cell-free culture supernatant, cytoplasm, periplasm, and membrane fraction. Pyrophosphatase, as the cytoplasm marker, was used to identify different cell fractions. TTH activity was not found in the cell-free culture supernatant, different from the TTH of *Acidianus hospitalis* [24]. The cytoplasm, periplasm, and membrane proteins presented 1.172 U, 0.391 U, and 0.059 U of TTH activity, respectively, which accounted for 72.3%, 24.1%, and 3.6% of the total TTH activity. The specific activity of the periplasmic fraction was 7.727 U/mg, of the cytoplasmic fraction was 0.526 U/mg, and 0.01 U/mg for the membrane at pH 3.0 and 0.009 U/mg for the membrane at pH 7.0 (Figure 3). The higher specific activity of the periplasmic fraction indicated a lower amount of TTH in the periplasm than in the cytoplasm, while the least TTH activity and the specific activity supported that TTH was not located on the membrane. The results indicated that TTH_Mc_ was primarily located in the cytoplasm with a lower amount present in the periplasm and barely any in the membrane (Appendix A).

### 2.5. Purification of the Tetrathionate Hydrolase from Whole Cell

Silver-stained SDS gels of different purification steps confirmed the presence of a TTH band at about 54 kDa in *M. cuprina* Ar-4 (Appendix A). The specific activity of TTH in the total cell extract of *M. cuprina* Ar-4 grown on 5 mmol/L K_2_S_4_O_6_ was found to be 0.72 U/mg. After four chromatographic purification steps including DEAE anion exchange column chromatography, hydrophobic chromatography, size exclusion chromatography, and Q-sepharose chromatography, the specific activity increased to 12 U/mg protein, representing a 17-fold purification with 1.2% recovery (Table 1 and Appendix A).

### 2.6. Tetrathionate Hydrolase of M. cuprina Is an Acid Resisting Thermophilic Enzyme

The enzyme activity of TTH_Mc_ was tested under different temperatures and pH conditions. As the temperature increased, the enzymatic efficiency increased almost linearly. The specific activity remained at approximately 30 U/mg at 95 °C (Figure 4A). However, the maximal and optimum temperature of TTH_Mc_ could not be determined due to the inability to reach a higher temperature range. The pH experiment showed that the maximum activity occurred at pH 6.0; the specific activity of TTH_Mc_ increased from 9.54 ± 0.17 U/mg at pH 1.0 to 39.44 ± 1.75 U/mg at pH 6.0, and then decreased at pH 7.0 (Figure 4B). An amount of 0.01 mol/L magnesium greatly stimulated the enzyme activity, while 0.01 mol/L calcium seriously inhibited the enzyme activity. Other divalent metal ions such as Fe^2+^, Zn^2+^, Mn^2+^, and Ni^2+^ had less effect on the specific activity of the enzyme (Table 2 and Appendix A).

The reaction process of tetrathionate catalyzed by TTH_Mc_ followed a Michaelis–Menten curve; the substrate hydrolysis velocity (V) was influenced by the enzyme and substrate concentration. When the enzyme concentrations were 1.15 μg/L and 2.30 μg/L, the half-lives of the substrates were 55.12 min and 27.36 min, respectively. The reaction rate constants corresponding to the half-life were K_1.15μg_ = 0.01258 mmol/L·min and K_2.30μg_ = 0.02533 mmol/L·min, respectively, and the enzyme concentration showed inversely proportional to the half-life (Appendix A). The kinetic parameters of TTH activity were measured at a temperature of 95 °C and pH of 6.0 with 1.15 µg TTH_Mc_. The apparent K_m_ value for tetrathionate was determined to be 0.35 mmol/L while the observed V_max_ was measured as 86.3 μmol/min (Figure 5).

### 2.7. Hydrolysis Products of Tetrathionate by Tetrathionate Hydrolase

An analysis of the enzymatic reaction mixture by HPLC revealed the presence of four peaks. Besides the peak of tetrathionate at a retention time of 9.5 min, thiosulfate presented at a retention time of 5 min (Figure 6). The retention times of pentathionate and hexathionate were 11.8 min and 21.6 min, respectively, which were also identified using the standards described by Miura and Kawaoi [29].

## 3. Discussion

Tetrathionate plays a significant role in the metabolism of acidophilic microorganisms, particularly in the RISC pathway as it serves both as a substrate and as a mesostate [31]. The location in cells and characters of the enzyme responsible for tetrathionate metabolism varies greatly among different species [12,15,22]. There is a lack of systematic studies on the tetrathionate hydrolase of acidothermophilic sulfur-oxidizing archaeon *Metallosphaera* genus, another member of the *Sulfolobaceae* family. In our study, a tetrathionate hydrolase was purified from *Metallosphaera cuprina* Ar-4, and the main objective of this investigation was to gain a comprehensive understanding of the tetrathionate hydrolase, determine its cellular localization, and characterize its biological properties.

### 3.1. TTH in Archaea and Bacteria Species

The TTH_Mc_ encoded by Mcup_1281 (PQQ-binding-like beta-propeller repeat protein) of *M. cuprina* is clustered to that of other *Metallophaera* spp. However, it exhibits differences with that of *Sulfolobus* spp., *Acidianus* spp., and *Sulfurisphaera* spp. It has been observed that the TTH cluster is conserved in *Metallophaera* [12]. Additionally, the region surrounding TTH contains an ABC transporter, a 4Fe-4S cluster domain-containing protein, and a vitamin epoxide reductase family protein (Appendix A). Tetrathionate hydrolases are widely found in sulfur-oxidizing archaea and bacteria, but the number of gene copies and protein location are not consistent across microorganisms [15]. There is no information available on the regulatory system that controls the tetrathionate intermediate pathway in sulfur-oxidizing archaea, such as the two-component system RsrS-RsrR found in *Acidithiobacillus* species [32]. We hypothesize that sulfur-oxidizing archaea have developed a specialized regulatory system to adapt to challenging environmental conditions.

The top-scoring BLAST entries of TTH_Mc_ at the Uniprot and NCBI databases (pairwise identity > 30%) were highly similar homologs found in acidophilic archaea and bacteria (Appendix A). These homologs belong to the PQQ family proteins and are likely to be authentic tetrathionate hydrolases. The proteins formed separate branches according to their respective species, indicating the conservation of tetrathionate hydrolases within the same species. However, there were some variations observed in tetrathionate hydrolases among different species. In addition, it was noted that the *Metallosphaera* spp. lacked the *tth*2 gene copies of the tetrathionate hydrolases found in *Acidianus* spp.

Unlike the tetrathionate hydrolase of *Ac. ambivalens*, TTH_Mc_ can also be detected in the cells of *M. cuprina* Ar-4 grown on S^0^ as the sole energy medium. The transcription level of TTH_Mc_ is higher in the cells grown on K_2_S_4_O_6_ than in the cells grown on S^0^. This suggests that the expression of tetrathionate hydrolase varies among different species and that tetrathionate metabolism in *M. cuprina* Ar-4 is TTH-dependent, similar to sulfur-oxidizing bacteria [33]. We also found the presence of Fe (II) in the broth from tetrathionate- and sulfur-grown *M. cuprina* Ar-4 cells through the addition of Fe (III) to the medium, and there is a possibility that it came from the contact of tetrathionate hydrolase with ferric reductase [34].

### 3.2. TTH Is Found in the Cytoplasm and Periplasm of K_2_S_4_O_6_ and S^0^-Grown Cells

It has been suggested that the TTH (TTH1) of *Ac. ambivlaens* is located in the pseudo-periplasmic space [15]. Additionally, it has been stated that the TTH of *Ac. hospitalis* was a secreted protein with a zipper-like shape [24]. However, we found that TTH_Mc_ is a soluble protein and it has the highest specific activity in periplasm and cytoplasm compared to other fractions. Moreover, the optimal pH of TTH_Mc_ is 6.0, unlike the TTH of other microorganisms. These results indicate that TTH_Mc_ is located in both the periplasm and the cytoplasm.

A series of purification strategies are based on the chemical properties of the protein, such as its hydrophobicity, charge property, and molecular weight. In a recent study, tetrathionate hydrolase was purified using four chromatographies [15]. The ion exchange column chromatography process with the NaCl concentrations of 150–300 mM resulted in similar charge properties of TTH_Mc_ as *Ac. ambivalens* TTH1 [15]. However, the hydrophobicity character differed between TTH1 and TTH_Mc_. TTH_Mc_ precipitated in 2 mol/L (NH_4_)_2_SO_4_. The specific activity consistently increased with each high number of purification steps, but the recovery yield was only 1.24%. This low yield may be attributed to purification steps, as increased steps often lead to a higher protein loss rate. Additionally, metal ions or cofactors like calcium may play a crucial role in stabilizing the protein structure [35].

### 3.3. Biochemical Properties of Tetrathionate Hydrolase

Some biochemical properties of the TTH_Mc_ of *M. cuprina* are similar to those of other acidophilic archaea [15]. The optimal temperature for this enzyme is above 95 °C, which is much higher than the enzymes from mesophilic species (optimum at 40–65 °C), even higher than the optimal growth temperature of *M. cuprina* [6]. By the protein sequence of the TTH_Mc_, we calculated that the isoelectric point (pI) of TTH_Mc_ is 4.70 by ExPASy ProtParam (https://web.expasy.org/protparam/ (accessed on 15 September 2024)), different from that of the TTH1 of *Ac. ambivlaens* which is 6.48 [15]. Nevertheless, TTH_Mc_ exhibited an optimal pH of 6.0, which is higher than the typical pH ranges [1.0–4.0] observed for TTHs from other sulfur-oxidizing bacteria and archaea [15]. This suggests that TTH_Mc_ is more effective in the intracellular environment. The molecular mass of the enzyme is approximately 57 kDa; however, no homodimer formation was observed. At pH 6.0 and 95 °C, the K_m_ value of TTH_Mc_ for tetrathionate is 0.35 mmol/L. Reported K_m_ values for TTH vary widely, suggesting differing substrate affinities across species. Among them, the TTH from *Thiobacillus ferrooxidans* (*Ac. ferrooxidans*) stands out with a K_m_ value of 50 μmol/L, appearing a higher substrate affinity [23].

### 3.4. Evolutionary Insights of TTHs Functionality

To delve deeper into the evolutionary history of TTH functionality, we constructed a protein sequence similarity network (SSN) using TTH_Mc_ as the query [36]. The analysis reveals that TTHs from the *Sulfolobaceae* family cluster closely with homologs predominantly from other acidophilic sulfur-/iron-oxidizing bacterial groups such as the *Acidithiobacillaceae*, *Alicyclobacillaceae*, *Sulfobacillus*, and *Leptospirillum*. These homologs form a dense cluster on the periphery of the entire SSN (Figure 7A) and maintain few connections with distantly related homologs of unknown function. This pattern supports the occurrence of the cross-domain horizontal gene transfer (HGT) of TTH, further corroborated by the unconserved genomic contexts within the *Sulfolobaceae* strains and the presence of nearby transposase elements (Appendix A). Previous studies have shown that an aspartate residue corresponding to Asp353 in TTH_Mc_ is crucial for the activity of the TTH from *Acidithiobacillus ferrooxidans* [19]. This aspartate residue is highly conserved within the TTH sub-cluster and is occasionally substituted by another acidic residue, glutamate (Figure 7B). This suggests that this aspartate residue initiates the reaction through protonating the S (alpha) atom of the tetrathionate [19].

Classical molecular dynamics were utilized to explore additional key interacting residues in TTH_Mc_ (Figure 7C). The analysis identified Arg460, Arg497, and Ala407 as the residues with the highest interaction rates. These residues likely serve as electrostatic stabilizers, forming hydrogen bonds with the terminal oxygen atoms adjacent to the S (alpha) atom of tetrathionate. Furthermore, Gln140, Thr139, Val136, and Arg132 of TTH_Mc_, positioned on a helix extending from the narrower propeller side, interact with the substrate via water-mediated bridges. Within the taxa from the TTH sub-SSN, Arg460 and Arg497 are frequently substituted by another basic residue (lysine) (Figure 7D,E). Additionally, other interacting residues lack aligned counterparts in remote sub-SSNs at equivalent positions, indicating the absence of corresponding structural regions (marked in gray) (Appendix A). Moreover, three methionine residues—Met172, Met238, and Met279—situated near the tetrathionate binding pocket have been identified as putative metal ligands in TTH_Mc_ (Appendix A). These residues are conserved within the sub-SSN of TTHs, though variations exist such as histidine replacing methionine in some instances. These findings suggest that the presence of these interacting residues likely enhances substrate binding and TTH activity across the TTH protein superfamily.

To further validate the functional significance of TTH-like proteins, future research should focus on functional assays and detailed experimental investigations. Such experiments will aid in confirming the role of these identified residues and their contributions to enzymatic activity. Overall, this study offers significant insights into the evolutionary dynamics of TTH_Mc_, setting the stage for a deeper understanding of tetrathionate hydrolase activity in related proteins.

## 4. Materials and Methods

### 4.1. Strains and Growth Conditions

*Metallosphaera cuprina* Ar-4^T^ was cultured at 65 °C in three different mediums derived by modifying Allen medium [6,14]. The BSM (Basic Salts Medium) contained (NH_4_)_2_SO_4_ 1.3 g, K_2_HPO_4_ 0.28 g, MgSO_4_·7H_2_O 0.25 g, CaCl_2_·2H_2_O 0.07 g, FeCl_3_·6H_2_O 0.02 g, MnCl_2_·4H_2_O 1.8 mg, Na_2_B_4_O_7_·10H_2_O 4.5 mg, ZnSO_4_·7H_2_O 0.22 mg, CuCl_2_·2H_2_O 0.05 mg, Na_2_MoO_4_·2H_2_O 0.03 mg, CoSO_4_·7H_2_O 0.01 mg, and VoSO_4_·2H_2_O 0.03 mg. The initial pH was adjusted to 3.5 with sterile 50% H_2_SO_4_. About 5 g sterile S^0^ (S^0^ medium) or 5 mmol/L K_2_S_4_O_6_ (TT medium) as sulfur source and 0.2% yeast extract were added in the BSM to promote cell growth. 2 g/L yeast extract was added in BSM for heterotrophic culture strains. Cell growth was monitored by optical density at 600 nm.

### 4.2. Sequence Analysis

The homologous amino acid sequences were searched in the UniProt and NCBI databases with Mcup_1281 as a probe. The multiple alignment (the homologous amino acid sequences from *Metallaspahera* spp., *Acidianus* spp., *Acidithiobacillus* spp., etc.) and a phylogenetic dendrogram were constructed using MAFFT version 7 with the L-INS-I algorithm, the BLOSUM 30 similarity matrix, a gap opening penalty of 1.53, and the offset value of 0.1. Using the ITOL (https://itol.embl.de/ (accessed on 1 August 2020)) aimed to modify the phylogenetic tree. The amino sequences of TTH from *Metallaspahera*, *Acidianus*, and *Acidithiobacillus* were compared and analyzed by ESPript 3.0 (http://espript.ibcp.fr/ESPript/cgi-bin/ESPript.cgi (accessed on 11 December 2020)).

The secondary structure of Mcup_1281 was predicted by SOPMA (https://npsa.lyon.inserm.fr/cgi-bin/npsa_automat.pl?page=/NPSA/npsa_sopma.html (accessed on 11 December 2020)). The prediction of signal sequences and transmembrane domain were performed with TMHMM and SignalP (http://www.cbs.dtu.dk/services/ (accessed on 11 December 2020)). The three-dimensional models of Mcup_1281 were predicted by AlphaFold2 [37] and the template model was quinoprotein alcohol dehydrogenase of *Pseudomonas putida* (PDB accession number 1kv9). The image of the prediction model was shown by Viewer Lite. The MD simulations for the protein–ligand complex were performed using the Desmond Molecular Dynamics System version 3.6 (D. E. Shaw Research, New York, NY, USA, 2008) with OPLS_2005 force field.

### 4.3. RNA Extraction from Cells and Transcription Analysis for mcup_1281

The cells grown on three different mediums containing 5 g/L S^0^, 5 mmol/L K_2_S_4_O_6_, and 2 g/L yeast extract, respectively, were harvested by 10,000× *g* centrifugation for 10 min. The total RNA of the cells was extracted with E.Z.N.A Total RNA Kit I (Omega). The quantity and quality of the RNA was measured by UV absorption (A260/A280 = 1.8–2.0) using NanoDrop One (Thermo Fisher Scientific, Waltham, MA, USA). The RNA was reverse transcribed by the HiFi-Script cDNA Synthesis Kit (CWBIO, Taizhou, China). The 16S rRNA gene of Ar-4^T^ was set as the reference gene and the *mcup_1281* as the target gene. The target DNA fragments used for synthesis were amplified with the primers Real-*TTH*-F (GCGGGGTTCACAAGTTTGAC) and Real-*TTH*-R (GCCTGTGCTGGATGATGT), while the reference gene fragments primers were Real-16S rRNA-F (TTGGGATCGAGGGCTGAAAC) and Real-16S rRNA-R (TCCCCTACGGCTACCTTGTT). LightCycler^®^ 480 System (Roche, Switzerland) was used for quantitative PCR and the program included pre-incubation at 95 °C 1 cycle, amplification, degeneration at 95 °C for 10 s, annealing at 55 °C for 20 s, and extension at 72 °C for 40 cycles; melting curve: 95 °C for 5 s, 55 °C for 1 min, 1 cycle.

### 4.4. Tetrathionate Hydrolases Purification from Whole Cells

The purification of tetrathionate hydrolases from whole cells was carried out by modifying the method described by Protze [15]. A total of 5 g of wet cells was collected from the TT medium or S^0^ medium. The cells were resuspended in 50 mL 40 mmol/L K_3_PO_3_ buffer (pH 7.0) with 10 mmol/L MgCl_2_, 25 μg/mL RNase, 10 μg/mL DNase I, and 1 mmol/L phenylmethanesulfonyl fluoride (PMSF). After adding 1 mm glass beads, the cells were disrupted three times through a homogenizer (Fast-prep-24, Santa Ana, MP, USA) with 6.0 m/s, 30 s. Cell debris and unbroken cells were discarded by centrifugation at 10,000× *g* and 4 °C for 15 min. The soluble fraction and membrane were separated by centrifugation at 140,000× *g* and 4 °C for 60 min. The supernatant was dialyzed against 40 mmol K_3_PO_3_ and 10 mmol/L MgCl_2_ (pH 7.0) at 4 °C for 16 h and the membrane was stored at −80 °C with phosphate-buffered saline (PBS) (pH 7.2). The dialyzed soluble fraction containing crude enzyme was applied on the DEAE Sepharose fast-flow column (5 mL column volume (CV); GE Healthcare) equilibrated with 40 mmol/L K_3_PO_3_ and 10 mmol/L MgCl_2_, pH 7.0. Proteins were eluted by a NaCl solution with a linear concentration gradient of 0–0.2 mol/L, 0.2–0.4 mol/L, 0.4–0.6 mol/L, 0.6–0.8 mol/L, and 0.8–1.0 mol/L, each for 5–10 CV. The eluent fraction showing activity of tetrathionate hydrolases was pooled and dialyzed against 50 mmol/L Tris/HCl (pH 8.0) supplemented by 1 mol/L (NH_4_)_2_SO_4_ for 16 h at 4 °C. After dialysis, the fraction was centrifugated at 40,000× *g* and 4 °C. The supernatant was next applied onto the Phenyl-Sepharose Fast-Flow column [5 mL column volume (CV); GE Healthcare]. The protein was eluted by 10 CV (NH_4_)_2_SO_4_ with a linear gradient from 1 mol/L to 0 mol/L. Fractions showing tetrathionate hydrolase activity were pooled and concentrated with Centriprep YM-30 centrifugal ultrafiltration tube (15 mL Millipore, Burlington, MA, USA) and then eluted by 40 mmol/L PBS (pH 7.5) with a flow rate of 0.5 mL/min through Superdex 200 size exclusion column (30 mm × 600 mm; GE Healthcare). The peak showing the tetrathionate hydrolase activity was applied on the Q-Sepharose-HP column [5 mL column; GE Healthcare]. Protein was eluted by NaCl with a linear gradient from 0.15 mol/L to 0.6 mol/L. Every purification fraction was detected with Silver-stained SDS-PAGE and HPLC enzyme assay to ensure tetrathionate hydrolase activity.

### 4.5. Cell Fractionation Separation

The cells of *M. cuprina* Ar-4 were grown to stationary phases and centrifugated at 10,000× *g* for 30 min. The cell-free supernatant was, in turn, filtered through filters with hole sizes of 2.2 μm, 0.8 μm, and 0.22 μm. The filtrate then was concentrated with Centriprep YM-30 centrifugal ultrafiltration tube (15 mL Millipore, Burlington, MA, USA) as the extracellular protein.

In total, 1 g wet cells was washed twice with 10 mmol/L Tris-HCl buffer solution and resuspended with 50 mL 10 mmol/L Tris-HCl buffer solution (pH 7.3), and then the equivalent volume of 10 mmol/L Tris-HCl buffer containing 40% sucrose was slowly added into and rocked slowly for 30 min at room temperature followed by centrifugation with 10,000× *g* at 4 °C. Then, the cells were quickly resuspended with 50 mL 5 mmol/L MgCl_2_ at 4 °C, and spheroplasts were removed by centrifugation at 10,000× *g* for 30 min. The supernatant was periplasmic protein [38].

Spheroplasts were washed three times with BSM and resuspended with 100 mmol/L Tris-HCl and disrupted by sonication using Vibra Cell, cell debris was removed by 10,000× *g* for 15 min, and the soluble fraction was centrifuged at 100,000× *g* for 60 min at 4 °C to separate the membrane (precipitate) and cytoplasmic (supernatant) fraction.

The membrane fraction was washed three times with distilled deionized water, resuspended in an elution buffer, and stayed on ice overnight. The formate buffer (pH 3.0) contained 0.5% (*v*/*v*) N-lauroylsarcosine, 1% (*v*/*v*) SDS, and 1% (*v*/*v*) Triton X-100 or 100 mmol/L. Tris-HCl buffer (pH 7.0) contained 0.5% (*v*/*v*) Triton X-100, and 5 mmol/L EDTA was used as elution buffer [38]. The fraction was centrifuged at 100,000× *g* for 60 min at 4 °C and the soluble fraction was concentrated with a Centriprep YM-30 centrifugal ultrafiltration tube (15 mL Millipore, Burlington, MA, USA) as the membrane protein fraction that was tested for TTH activity.

### 4.6. Enzyme Characteristic Assay

The tetrathionate hydrolase activity of different fractions was determined by two methods.

#### 4.6.1. Continuous Method of Enzyme Activity Assay

The continuous assay is based on the principle that tetrathionate hydrolases hydrolyze K_2_S_4_O_6_ to produce a series of sulfur-containing compounds that increase absorbance at 290 nm [29]. The reaction mixture (1 mL, pH 3.5) contained 40 mmol/L K_3_PO_3_, 1 mmol/L K_2_S_4_O_6_, and 25–100 μL cell lysate or distillation water as control. Enzyme activity was detected by a full-wavelength scan which found that the absorption value of the mixture increased from 250 nm to 430 nm. This could represent that tetrathionate hydrolase hydrolyzes K_2_S_4_O_6_. In order to ensure the sensitivity and practicality of the measurement, the change rate of absorbance at 290 nm was selected to define the catalytic efficiency of tetrathionate hydrolases [29]. One unit enzyme activity was defined as the ΔA_290_ min^−1^.

Pyrophosphatase activity was detected with the method of Richter and Schäfer to assess the purity of each fraction [15,39]. A total of 1 μg protein from each fraction was incubated with a reaction buffer containing 40 mM acetate, 40 mM imidazole, 40 mM Tris, 2.5 mM MgC1_2_, and 0.2 mM Ppi (Na_4_ P_2_ O_7_) for 30 min at 60 °C. The concentration of Ppi was detected by a Pyrophosphate Assay Kit (MAK168, Sigma-Aldrich, Burlington, MA, USA). A unit pyrophosphatase activity was defined as the amount of protein required for the hydrolysis of 1 µmol/L of Ppi in one minute. The protein concentration of different fractions of cells was detected by the BCA assay.

#### 4.6.2. Discontinuous Method of Enzyme Activity Assay

For understanding the products from tetrathionate hydrolyzation, we used the HPLC assay method described by Miura and Kawaoi [29] to detect the sulfur-containing intermediates and quantify the activity of purified tetrathionate hydrolases. The K_2_S_4_O_6_ used in the study was purchased from Merck KGaA (Darmstadt, Germany) and its purity was over 98%. Na_2_S_2_O_3_ was purchased from Sinopharm (Shanghai, China), and its purity was over 99%. (1) The determination of the optimal temperature of TTH was proceeded in 1 mL 50 mmol/L formate buffer (pH 3.0) containing 1–2 mmol/L K_2_S_4_O_6_ and 1 mol/L (NH_4_)_2_SO_4_ (Sigma-Aldrich, Burlington, MA, USA). The reaction was started by adding 25 μL protein and hatched in a water bath from 25 °C to 95 °C with an interval of 10 °C. (2) The determination of the optimal pH of TTH carried in the same solution system of the temperature test and we set pH at 1–7 with an interval of 1.0, and hatched at 65 °C. (3) Vmax and Km were detected and calculated at 1.15 µg TTH with different tetrathionate concentrations. To stop the enzyme reaction, the sample was rapidly frozen in liquid nitrogen for 3 min and then boiled in boiling water for 5 min, repeated three times. The mixture was filtered by a 0.22 μm filter of polyethersulfone (SLGP033R, Millipore, Burlington, MA, USA). The contents of the chemical compounds were determined using an Agilent 1290 infinity II (Agilent, Waldbronn, Germany) consisting of a binary pump system and auto-sampler. A chromatography separation was achieved with an Agilent Eclipse Plus C18 column (4.6 mm × 150 mm, 5 µm) and maintained at 30 °C during the run. The mobile phase was acetonitrile (Thermo Fisher, USA)/water (20:80 *v*/*v*) with 6 mmol/L Tetrapropylammonium hydroxide (TPA, Sigma-Aldrich, USA) with adjusted pH as 5.0 by acetic acid. The flow rate was 0.8 mL·min^−1^ and the UV-detector was set at 230 nm. A unit enzyme activity of tetrathionate hydrolase was defined as the amount of protein required for hydrolysis of 1 µmol/L of S_4_O_6_^2−^ in one minute. The specific enzyme activity was the activity of per mg TTH.

### 4.7. Effects of Bivalent Ions on Enzyme Activity

The series of bivalent metal ions almost were metal chlorides, including magnesium chloride, zinc chloride, ferric chloride, nickel chloride, manganese chloride, and calcium chloride. The effects of metal ions on TTH_Mc_ activity were detected by the discontinuous methods of enzyme activity assay with 0.01 mol/L metal ion at 95 °C and pH 6.0.

### 4.8. Statistics

Statistics were performed using Prism 8 (Graphpad). All the quantification tests were independently performed for at least three replications. Data were represented as the mean. Significance was determined using a one-way ANOVA analysis, * *p* < 0.05, ** *p* < 0.01, *** *p* < 0.001, and **** *p* < 0.0001 throughout the manuscript.

## Figures and Tables

**Figure 1 ijms-26-01338-f001:**
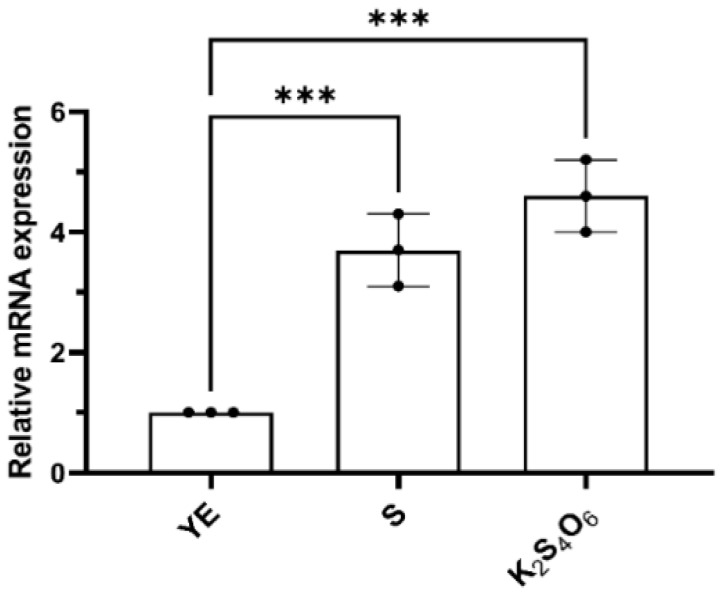
The transcriptional expression of *mcup_1281* in different cultures. YE, S, and K_2_S_4_O_6_ represent that the *M. cuprina* Ar-4 was cultivated with yeast extraction (YE), sulfur powder (S^0^), or K_2_S_4_O_6_ as the energy source, respectively. *16S rRNA* was used as the reference gene and *mcup_1281* as the target gene. The error bars represent the SD of *n* = 3 independent measurements. Significance was determined using a one-way ANOVA analysis, *** *p* < 0.001.

**Figure 2 ijms-26-01338-f002:**
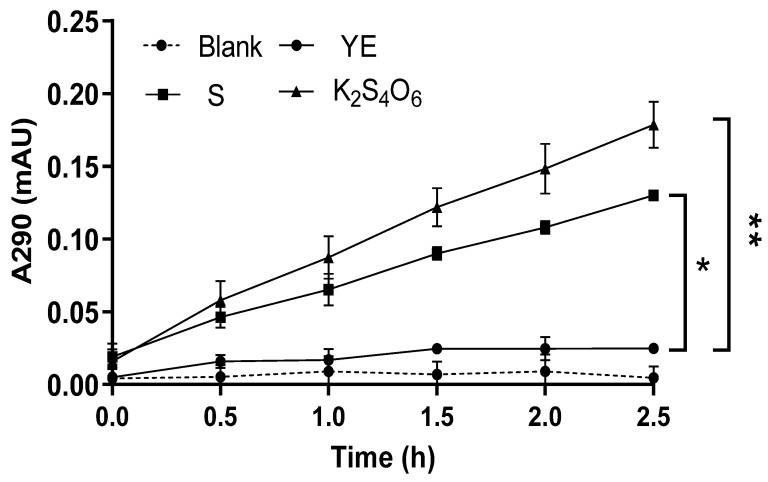
Activity of TTHs for cell lysis solution in different cultures analyzed by the continuous method. YE, S, and K_2_S_4_O_6_ represent that the *M. cuprina* Ar-4 was cultivated with yeast extraction (YE), sulfur powder (S^0^), or K_2_S_4_O_6_ as the energy source, respectively. The error bars represent the SD of *n* = 3 independent measurements. Significance was determined using a one-way ANOVA analysis, * *p* < 0.05, ** *p* < 0.01.

**Figure 3 ijms-26-01338-f003:**
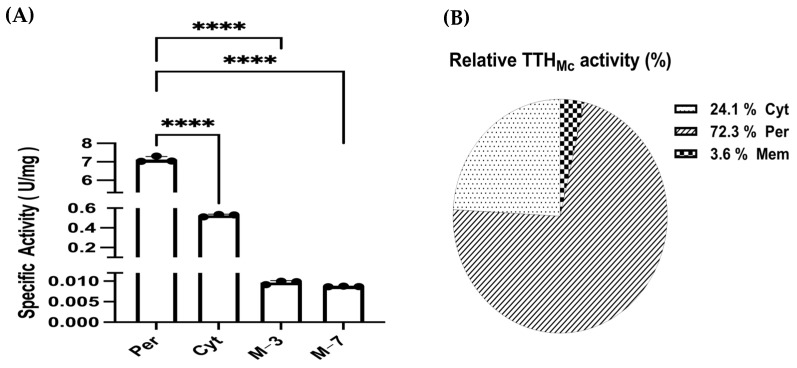
The TTH specific activity (**A**) and relative activity (**B**) in differential cell fractions of the whole cell. Per, periplasmic space; Cyt, cytoplasm; M−3, membrane (pH 3.0); M−7, membrane (pH 7.0). Three dots reprent three independent experiment values. The error bars represent the SD of *n* = 3 independent measurements. Significance was determined using a one-way ANOVA analysis, **** *p* < 0.0001. TTH activity was measured by the discontinuous method.

**Figure 4 ijms-26-01338-f004:**
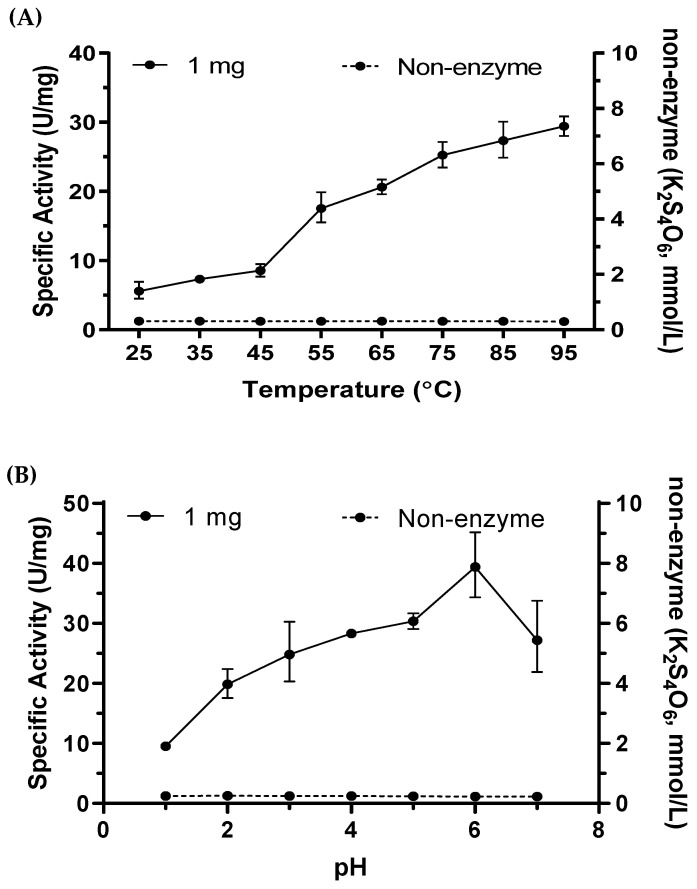
Temperature and pH profiles of TTH_Mc_ activity. (**A**) Temperature profile of TTH activity from 25 to 95 °C; (**B**) pH profile of TTH activity from 1 to 7. Error bars represent SD of *n* = 3 independent measurements. TTH activity was measured by discontinuous method.

**Figure 5 ijms-26-01338-f005:**
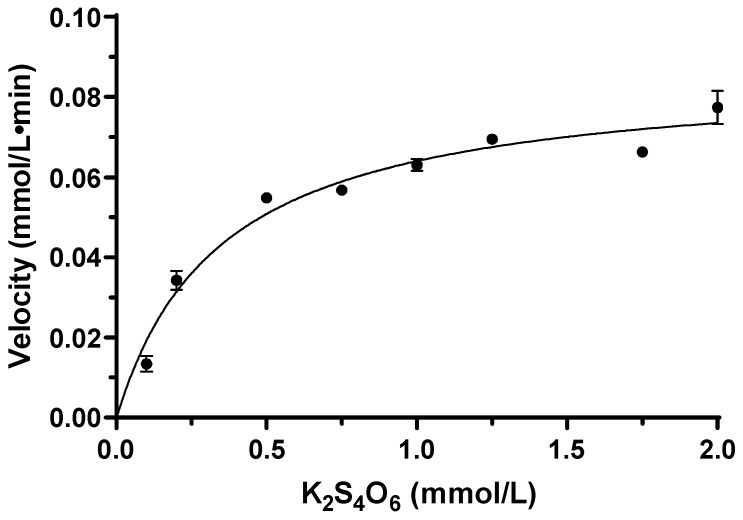
Relationship between TTH_Mc_ enzymatic reaction rate and K_2_S_4_O_6_ concentration. Error bars represent SD of *n* = 3 independent measurements. TTH activity was measured by discontinuous method.

**Figure 6 ijms-26-01338-f006:**
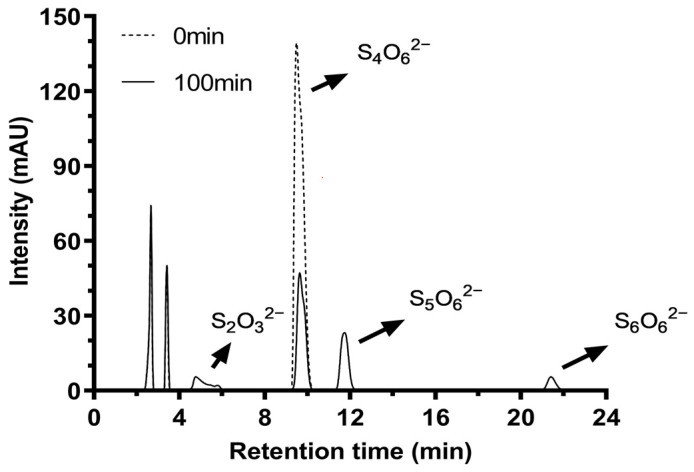
TTH_Mc_ products from *M. cuprina* Ar-4 analyzed by the HPLC method. The products were analyzed following the enzyme addition (Dotted line) and after 100 min incubation (Solid line).

**Figure 7 ijms-26-01338-f007:**
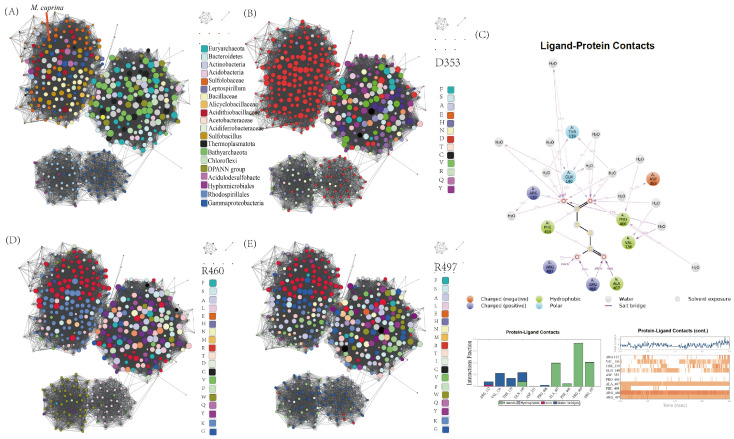
Sequence similarity network (SSN) built with TTH_Mc_ as the query. (**A**) SSN built with TTH_Mc_ as the query colored by taxonomy. (**B**) SSN built with TTH_Mc_ as the query colored by the type of the residues on the position of TTH_Mc_ Asp353. (**C**) Schematic of detailed ligand atom interactions with the protein residues. SSN built with TTH_Mc_ as the query colored by the type of the residues on the position of TTH_Mc_ (**D**) R460; (**E**) R497.

**Table 1 ijms-26-01338-t001:** Purification of the TTH from *M. cuprina* Ar-4. TTH activity was measured by the discontinuous method.

Fraction	Total Protein (mg) ^a^	Total Activity (U) ^b^	Specific Activity (U/mg) ^c^	Yield (%)	Fold
Cell extract	79 ± 5	57 ± 1	0.7 ± 0.2	100.0	1.0
DEAE	36 ± 1	50 ± 6	1.4 ± 0.1	88.5	2.0
HIC	1.2 ± 0.04	2.6 ± 0.2	2.2 ± 0.1	4.6	3.1
Size exclusion	0.33 ± 0.01	1.2 ± 0.2	3.8 ± 0.1	2.2	5.4
Q-sepharose	0.058 ± 0.001	0.7 ± 0.04	12 ± 1	1.2	17

^a–c^ represent the SD of *n* = 3 independent measurements.

**Table 2 ijms-26-01338-t002:** Effects of bivalent ions on enzyme activity. TTH activity was measured by discontinuous method.

Metal Ions	Concentration (mol/L)	Specific Activity (U/mg)	Residual Activity (%)
Control	-	6.57 ± 0.12	-
Mg^2+^	0.01	7.95 ± 0.04	121
Zn^2+^	0.01	6.55 ± 0.12	94.8
Fe^2+^	0.01	6.23 ± 0.09	94.5
Ni^2+^	0.01	6.10 ± 0.00	92.8
Mn^2+^	0.01	6.23 ± 0.09	87.7
Ca^2+^	0.01	3.93 ± 0.05	59.8

## Data Availability

All the data in this study are available within the article, Appendix A, and/or from the corresponding author upon reasonable request.

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
