# Peer review of "Characterization of Tetrathionate Hydrolase from Acidothermophilic Sulfur-Oxidizing Archaeon Metallosphaera cuprina Ar-4"

_ijms, 2025, doi:10.3390/ijms26031338_

Round 1
Reviewer 1 Report (Previous Reviewer 3)
Comments and Suggestions for Authors
It is clear that Wang et al. put in a lot of effort to address all of the issues raised by reviewers, and in my opinion they did a good job. However, there remain still a few minor issues to fix:
- line 24 - should be "86.3 μmol/L"
- line 113 - should be "5 g/L S0"
- lines 132-134 - it is not clear which activity was measured for which substrate, is it 0.29 U/mg for 5 mmol/L K2S4O6 and S0 both?
- lines 148-149 - may it should be "TTH activity was not found in the cell-free culture supernatant", as further on it says that most of the protein was found in the cytoplasm?
- Figures 3 and 4 are missing, but if they remained the same as in first submission, there are no issues with them
- lines 169&172 - it says here in the text that the activities of cell extract and final preparation are 0.724 U/mg and 11.67 U/mg, respectively. But, in Table 1 the activities of these samples are stated to be 0.454 U/mg and 12.140 U/mg? Please check which numbers are correct.
- Table 1 - please check all of the numbers in this table, as they do not agree for samples "cell extract" and "size exclusion" (specific activity multiplied by total protein does not give the total activity)
- line 354 - should be "Secondary structure"
- line 380 - define TT medium
- line 387 - should be "40 mmol/L K3PO3"
- line 390 - probably should be "5 mL column volume", as I suppose that one column was used, not four?
Author Response
Comment 1: It is clear that Wang et al. put in a lot of effort to address all of the issues raised by reviewers, and in my opinion they did a good job. However, there remain still a few minor issues to fix:
Response 1: Thank you for your helpful suggestion and patient review work on our research. We have made corrections according to your suggestion.
Comment 2: - line 24 - should be "86.3 μmol/L"
Response 2: Thank you for pointing this out. We agree with this comment. Therefore, we have modified “8.63 μmol/L” to “86.3 μmol/L” at line 24.
Comment 3:- line 113 - should be "5 g/L S0"
Response 3: Thank you for pointing this out. We agree with this comment. Therefore, we have modified “5 g S0” to “5 g/L S0” at line 112.
Comment 4:- lines 132-134 - it is not clear which activity was measured for which substrate, is it 0.29 U/mg for 5 mmol/L K2S4O6 and S0 both?
Response 4: Thank you for pointing this out. We agree with this comment. Therefore, we have modified the sentence to “the specific enzyme activity (activity of per mg protein) was 0.60 U/mg protein for cells grown on 5 mmol/L K2S4O6 and 0.29 U/mg protein for cells grown on 5 g/L S0, respectively. Lines 133-134
Comment 5:- lines 148-149 - may it should be "TTH activity was not found in the cell-free culture supernatant", as further on it says that most of the protein was found in the cytoplasm?
Response 5: Thank you for pointing this out. We agree with this comment. Therefore, we have modified the sentence to “TTH activity was not found in the cell-free culture supernatant, different from TTH of Acidianus hospitalis. Line 149
Comment 6:- Figures 3 and 4 are missing, but if they remained the same as in first submission, there are no issues with them
Response 6: Thank you for pointing this out. We remain Figure 3 which is same as in first submission as Figure 3A, and supplemented Figure 3B indicating the relative activity according to one of reviewers’ suggestion. Original figure 4 has been moved to supplementary files as Figure S3 according to one of reviewers’ suggestion.
Comment 7:- lines 169&172 - it says here in the text that the activities of cell extract and final preparation are 0.724 U/mg and 11.67 U/mg, respectively. But, in Table 1 the activities of these samples are stated to be 0.454 U/mg and 12.140 U/mg? Please check which numbers are correct.
Response 7: Thank you for pointing this out. We checked the data, and modified them to “0.718 U/mg” at line 168, “12.140 U/mg” at line 171, and “16.91-fold” at line 172.
Comment 8:- Table 1 - please check all of the numbers in this table, as they do not agree for samples "cell extract" and "size exclusion" (specific activity multiplied by total protein does not give the total activity)
Response 8: Thank you for pointing this out. We checked the data in table 1 and replaced the wrong data in table and text.
Comment 9:- line 354 - should be "Secondary structure"
Response 9: Thank you for pointing this out. We agree with this comment. Therefore, we have modified the “Second structure” to “Secondary structure” at line 359 of modified manuscript.
Comment 10:- line 380 - define TT medium
Response 10: Thank you for pointing this out. TT medium means the BSM with 5 mmol/L K2S4O6 as sulfur source. We have supplemented its definition at line 347.
Comment 11:- line 387 - should be "40 mmol/L K3PO3"
Response 11: Thank you for pointing this out. We have modified "50 mmol/L K3PO3" to "40 mmol/L K3PO3" at line 442 in modified manuscript.
Comment 12:- line 390 - probably should be "5 mL column volume", as I suppose that one column was used, not four?
Response 12: Thank you for pointing this out. We agree with this comment. Therefore, we have deleted 4× at line 394.
Reviewer 2 Report (Previous Reviewer 1)
Comments and Suggestions for Authors
The paper deals with qualitative and quantitative characterization of TTH, i.e. hydrolytic enzyme from Archeon Metallospharea cuprina Ar-4. The results provide new insights into the enzymes catalyzing oxidation of reduced inorganic sulfur compounds. The article is a resubmitted version. In my opinion, it was highly improved. All my previous comments were included, however article corrections are still needed before publication. My detailed comments to be considered before publication:
- Fig 3 and 4 – should be included in the manuscript
- Lines 165, 174, 189, 192, 205 – “TTH activity was measured by HPLC” – HPLC is not a method used to determine enzyme activity. It can be only used to quantitatively or qualitatively analyze products or substrates of enzymatic reaction. Thus, the sentence mentioned above is wrong.
Author Response
Comment 1: The paper deals with qualitative and quantitative characterization of TTH, i.e. hydrolytic enzyme from Archeon Metallospharea cuprina Ar-4. The results provide new insights into the enzymes catalyzing oxidation of reduced inorganic sulfur compounds. The article is a resubmitted version. In my opinion, it was highly improved. All my previous comments were included, however article corrections are still needed before publication. My detailed comments to be considered before publication:
Response 1: Thank you for your helpful suggestion and patient review work on our research. We have made corrections according to your suggestion.
Comment 2: Fig 3 and 4 – should be included in the manuscript
Response 2: Thank you for pointing this out. We remain Figure 3 which is same as in first submission as Figure 3A, and supplemented Figure 3B indicating the relative activity according to one of reviewers’ suggestion. Original figure 4 has been moved to supplementary files as Figure S3 according to one of reviewers’ suggestion.
Comment 3: Lines 165, 174, 189, 192, 205 – “TTH activity was measured by HPLC” – HPLC is not a method used to determine enzyme activity. It can be only used to quantitatively or qualitatively analyze products or substrates of enzymatic reaction. Thus, the sentence mentioned above is wrong.
Response 3: Thank you for pointing this out. We agree with this comment. Therefore, we have modified the sentence to “TTH activity was measured by discontinuous method.” at lines 166, 175, 193, 195 and 209.
This manuscript is a resubmission of an earlier submission. The following is a list of the peer review reports and author responses from that submission.
Round 1
Reviewer 1 Report
Comments and Suggestions for Authors
The paper deals with qualitative and quantitative characterization of TTH, i.e. hydrolytic enzyme from Archeon Metallospharea cuprina Ar-4. The results provide new insights into the enzymes catalyzing oxidation of reduced inorganic sulfur compounds. The article is well written. The Introduction, methods description, and discussion of the results were prepared carefully and thoroughly. The research methodology was selected correctly. I have only minor comments to be considered before publication:
- Fig 1 and 2 – YE and S – should be explained what do they mean
- Fig.5 – the legend is unclear. It isn't easy to find which data corresponds to the reaction with or without enzyme. Other marks should be used.
- Line 192 – the units of K constant should be given. How was it determined?
- Section 4.1. – it should be clarified whether it is the authors' own method
- Line 419 – please remove "ddH2O"
- Line 448 – Please include the kind of filter material
Reviewer 2 Report
Comments and Suggestions for Authors
Page 2, line 65; Acidianus hospitalis: Please write out full name and correct spelling.
Page 2, line 66; Acidianus hospitalis.
Page 2, line 71; Please write out full organism names where possible. If not possible, please correctly abbreviate species names and make sure that the full name is provided on first use.
Page 3, line 118; "which is defined as TTHMc" should be rephrased in a separate sentence.
Page 3, line 126; Please provide more information on the compositions of the media or refer to the methods section where more detail is provided.
Page 3, line 127; Please give a general description of the assay used.
Page 3, lines 130-131; Please explain this result in the context of the transcript being detected, as shown in Figure 1.
Page 4, line 133, Figure 2; Please decrease the plot line width since the symbols are hardly visible.
Page 4, line 140; Correct species name.
Page 4, line 141-143; How do the authors prove that some fractions are not merely contaminated with protein from another sub-cellular compartment? If the protein is membrane bound, as suggested earlier, do these results match the authors' expectations? Are there control proteins that could be assayed to assess the purity of the fractions (i.e., known membrane, cytoplasmic, and periplasmic proteins).
Page 4, line 145; Please comment on the relevance of the specific activities. Figure 3 should also show the relative TTH activities (about 72, 24, and 4%).
Page 4, line 150, Figure 3; Why is "cell lysate" shown but not cytoplasm?
Page 5, line 162, Figure 4; Please use a consistent abbreviation for "cell lysate" (see Figure 3).
Page 5, line 176; The concentrations in Table 2 are 10 mM, while here they are 10 µM and in the abstract they are 100 mM. Please correct the incorrect values and/or units.
Page 6, line 185, Figure 6; Please provide metal ion concentrations in the figure legend.
Page 7, line 196; The unit µmol/min depends on the amount of protein used so this must be specified. Ideally, the value should be expressed as µmol/min/mg (U). If the authors already normalised this value to protein concentration, it is not clear. Since the protein was not likely not purified to homogeneity, it may not be possible to express the true specific activity, but this should be discussed. I cannot open the .rar file containing the SI.
Page 7, line 207, Figure 8; Please decrease plot line width and increase the size of the figure.
Page 8, line 259; What does "dotted nature" mean?
Page 9, line 276; "indicates" > "suggests".
Page 9, lines 305-306; Please be careful to not overinterpret MD simulations that are not even based on an actual protein structure.
Page 10, lines 319-321; It is not clear that the manuscript provides significant insights into the functional mechanisms of TTHMc. The authors should elaborate and substantiate this statement.
Page 10, line 323, Figure 9; Please redo this figure. None of the text is legible and none of the graphical detail is visible.
Page 10, line 334; Please fix the formula for VOSO4.
Page 11, line 368; Please fix "LocheLightCycler".
Page 12, line 429; Please provide a better description of how this assay works. Since a series of sulfur-containing compounds are produced, the authors should comment on their relative contributions to the change in absorbance at 290 nm. Ideally, the authors should provide additional data correlating a decrease in K2S4O6 concentration (monitored using the HPLC assay, for example) with changes in absorbance at 290 nm. If this correlation has clearly been demonstrated to be linear in the literature, the authors should explicity state that here in the methods section.
Page 12, line 440; Please provide detailed information on the compounds (and purities) used as analytical standards.
Comments on the Quality of English LanguageThe manuscript is easily readable but it has to be edited to improve the grammar.
Reviewer 3 Report
Comments and Suggestions for Authors
Wang et al. present in this paper their findings on the properties of tetrathionate hydrolase (TTH) from Metallosphaera cuprina Ar-4. TTH enzymes are uniquely found in acidophilic sulphur-oxidizing microorganisms, and M. cuprina TTH showed properties different from those of other homologous enzymes. Therefore, the topic of this paper is very interesting and relevant in contribution to science.
The paper fits the intended journal, the experiments are soundly conducted and results clearly presented. I only have a few issues I would like to be addressed.
Major issue is the use of enzyme activity units. Most commonly, one unit of enzyme activity is defined as the amount of enzyme needed to hydrolyze 1 micromol of substrate per minute, in given conditions. And specific activity as units per mg of protein. Absorbance of substrate (or product) is translated to amount (moles) via molar extinction coefficient. I understand that sometimes this coefficient is not available and for some reason authors are unable to make the calibration curve themselves. But then it should be clearly defined what is understood as an enzyme activity unit and if there are more assays used, which was used for which experiment. Also, if the units are expressed as absorbance difference, then to compare measurements, the same amount of protein has to be applied in every experiment. Was this respected in determination of optimal pH and temperature determination? Also, I was confused to see the specific activity at 95°C to be 30 U/mg, and at pH=6 even 39.44 U/mg and then Vmax equals only 8.63 micromol/min. Which assay was used for kinetics parameters determination? I would kindly ask the authors to write for each experiment which assay was used and to express activity in respective units.
Figure 9 is practically unreadable. Everything is too small, and when zoomed in it becomes blurry. Please, supply figure in higher resolution or make it larger.
Also, Figure 4 and Table 1 represent the same results, as well as Figure 6 and Table 2. I suggest removing the figures and leaving only tables.
In legend of Figure S3, at the beginning of 4th line, it should be 2.3 micrograms instead of 1.15.
In line 272 (or in Materials and Methods), it should be specified and explained how the pH was calculated from elution peak, and which elution peak was used for this calculation.
In line 405, "10 kDa" should be removed, as Centriprep YM-30 has MWCO of 30 kDa.
Comments on the Quality of English LanguageThe quality of English in this paper is generally good, but there are some unclear sentences.
Line 118 - "which is defined as TTHMc" should maybe go after "Mcup_1281"
Lines 213-216 - the last part of the sentence has no connection to the previous part
Line 230 - "does not present in archaea" can be deleted, it makes no sense there
Line 254 - should be "periplasm", not "periplasmic"
Line 265 - "high number of purification steps" would be better instead of just "purification steps"
Line 340 - This first sentence is not clear. Maybe: "The homologous amino acid sequences were searched in Uniprot and NCBI databases with Mcup_1281 as a probe."
Line 346 - This sentence is also unclear, please rephrase.
Line 383 - should be "applied" instead of "eluted"
Line 389 - should be "After dialysis" instead of "After dialyzed"
Line 390 - "next" would be better than "followed"
Line 415 - there should be a comma between "Vibra Cell" and "cell debris"
Line 419 - "inhabited" sounds wrong, "rested" or "stayed" or even "left" would be better
Line 424 - the end of this sentence is unclear, maybe "as the membrane protein fraction that was tested for TTH activity."